# Measuring Taiwanese Mandarin Language Understanding

**Po-Heng Chen, Sijia Cheng, Wei-Lin Chen, Yen-Ting Lin, Yun-Nung Chen**
National Taiwan University, Taipei, Taiwan
{ytl, yvchen}@ieee.org

## Abstract

The evaluation of large language models (LLMs) has drawn substantial attention in the field recently. This work focuses on evaluating LLMs in a Chinese context, specifically, for Traditional Chinese which has been largely underrepresented in existing benchmarks. We present TMLU, a comprehensive evaluation suit tailored for assessing the advanced knowledge and reasoning capability in LLMs, under the context of Taiwanese Mandarin. TMLU consists of an array of 37 subjects across social science, STEM, humanities, Taiwan-specific content, and others, ranging from middle school to professional levels. In addition, we curate chain-of-thought-like few-shot explanations for each subject to facilitate the evaluation of complex reasoning skills. To establish a comprehensive baseline, we conduct extensive experiments and analysis on 24 advanced LLMs. The results suggest that Chinese open-weight models demonstrate inferior performance comparing to multilingual proprietary ones, and open-weight models tailored for Taiwanese Mandarin lag behind the Simplified-Chinese counterparts. The findings indicate great headrooms for improvement, and emphasize the goal of TMLU to foster the development of localized Taiwanese-Mandarin LLMs. We release the benchmark and evaluation scripts for the community to promote future research.[1]

## 1 Introduction

The emergence of large language models (LLMs) has revolutionized the field of natural language processing (NLP). As the core that drives the trajectory of AI development, evaluation stands a crucial role in the era of LLMs. Conventional NLP benchmarks (Wang et al., 2018; 2019) have been widely adopted to evaluate natural language understanding (NLU) abilities in LMs. However, the applicability of these benchmarks has decreased as the ever larger models are demonstrating human-level performance, leaving little headroom for research developments (Hendrycks et al., 2020; Goyal et al., 2022; Liu et al., 2023).

Towards appropriately benchmarking LLMs, probing advanced world knowledge and measuring complex reasoning capabilities are the main focus LLM evalution in recent days (Clark et al., 2018; Hendrycks et al., 2021; Cobbe et al., 2021; Wang et al., 2023). Additionally, evaluation benchmarks for languages beyond English have also been introduced (Li et al., 2023; Huang et al., 2023), in parallel to the rise of multilingual LLMs (Le Scao et al., 2022; Muennighoff et al., 2022; Wei et al., 2023) and LLMs that are optimized for different regional languages (Lee et al.; Nguyen et al., 2023).

Specifically, a number of benchmarks (Huang et al., 2023; Li et al., 2023; Xu et al., 2023) have been proposed for assessing Chinese LLMs (Zeng et al., 2022; Yang et al., 2023; Bai et al., 2023; AI et al., 2024). However, these benchmarks and models are all developed under the context of Simplified Chinese used in Mainland China. On the other hand, Traditional Chinese, often referred to as Traditional Mandarin, has been notably underrepresented. Predominantly used in Taiwan, Hong Kong, and Macao, Traditional Mandarin carries different language usage between countries or locations. Specifically, Taiwanese Mandarin

---

[1] https://github.com/MiuLab/TMLU

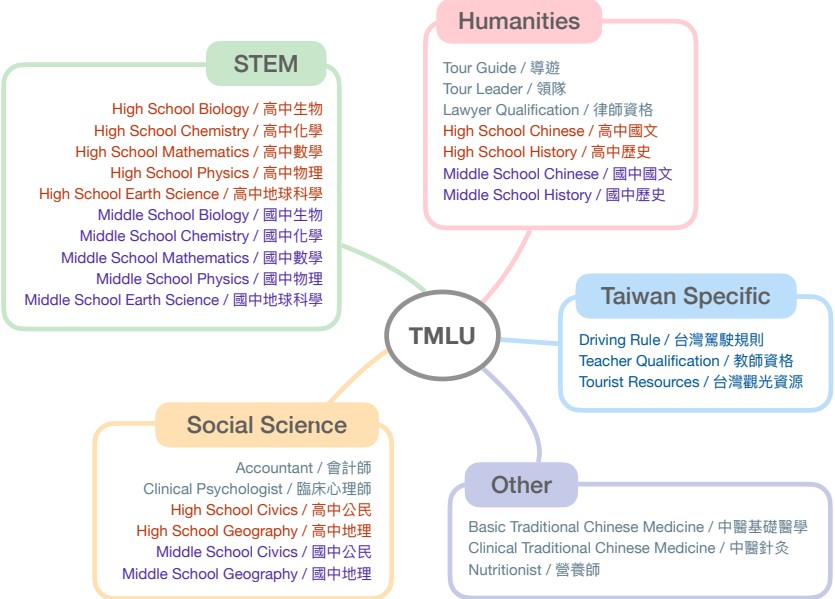

Figure 1: An overview of our proposed TMLU benchmark. TMLU consists of 37 subjects across middle school, high school and professional levels. In addition, TMLU includes a set of Taiwan specific questions in which answers are unique to Taiwanese culture.

posses linguistic nuances, cultural intricacies, and a different written form which diverge from Chinese used in China and present unique challenges (Wang & Tang, 2021; Lin & Chen, 2023). For example, the word "土豆 (Tudou)" typically means "potato" in Mainland China, while it refers to "peanut" in Taiwan (Zhou & Zhou, 2019). Consequently, the need of building benchmarks organically in Taiwanese Mandarin is necessitated to facilitate the development of localized, Taiwanese-Mandarin LLMs.

In this work, we bridge the gap and present TMLU — a comprehensive evaluation suit tailored for assessing advanced knowledge and reasoning capability in LLMs under Taiwanese-Mandarin context. TMLU consists of a broad spectrum of 37 subjects, spanning social science, STEM, humanities, Taiwan-specific content, and others, from middle school to professional level. Including high school and college entrance exams, civil service exams, and other localized knowledge in Taiwan. Furthermore, TMLU includes manually curated, Chain-of-Thought-inspired (Wei et al., 2022) explanations to facilitate the evaluation of reasoning capability in LLMs. See Figure 1 for an overview of TMLU.

To establish a comprehensive baseline, we evaluate 24 advanced LLMs, including both open-weight and proprietary models, on TMLU. The experimental results indicate that, in general, proprietary black-box LLMs with multilingual capability outperform open-weight models developed by Chinese communities. Also, open-weight models tailored for Taiwanese Mandarin demonstrate suboptimal scores comparing to models build predominantly in Simplified-Chinese background. Overall, the findings suggest that there still exist ample rooms for improvement, underscoring our goal to facilitate the progress of Taiwanese-Mandarin LLMs. In addition, we present a number of analysis and conduct data contamination test to further validate the reliability and robustness of TMLU.

## 2   Related Work

Prior to the bloom of LLMs, standard NLP benchmarks such as GLUE (Wang et al., 2018) and SuperGLUE (Wang et al., 2019) have been widely adopted to evaluate linguistic proficiency and natural language understanding (NLU) abilities in LMs. However, the practicality of these benchmarks in the era of LLMs have saturated as new models are demonstrating

|  | TC-Eval | TMMLU-plus | TMLU (ours) |
|---|---|---|---|
| Robustness | medium | low | high |
| Localization | low | high | high |
| Coverage | low | high | medium |
| Transparency | high | low | high |
| Standardization | high | medium | high |

Table 1: A comparison of existing Taiwanese Mandarin evaluation benchmarks and our proposed TMLU. Notably, TMLU is superior to TC-Eval in the degree of localization, and is more robust to dataset contamination compared to TMMLU-plus. We present a detailed discussion in Section 2.

superhuman performance (Goyal et al., 2022; Liu et al., 2023). In light of this, benchmarks which focus on assessing the world knowledge and complex reasoning abilities in LLMs have been introduced recently(Hendrycks et al., 2020; Srivastava et al., 2022; Hendrycks et al., 2021; Suzgun et al., 2022; Liang et al., 2022). The MMLU benchmark (Hendrycks et al., 2020) covers an array of subjects across STEM and social sciences, from elementary to expert levels, for comprehensive multitask evaluation. The BIG-bench benchmark (Srivastava et al., 2022) contains a large collection of diverse tasks that are designed to be beyond the current ability of LLMs. However, these benchmarks are primarily constructed in English, which limit their usage in assessing LLMs developed in the context of other languages.

To this end, several benchmarks have been proposed to facilitate evaluation of Chinese LLMs. C-Eval (Huang et al., 2023) presents one of the first comprehensive evaluation suits built to assess LLMs' advanced knowledge and reasoning capabilities in Chinese context, with subjects across social science, STEM, and humanities. Another concurrent benchmark, CMMLU (Li et al., 2023), shares a similar vision and includes questions relevant to Chinese users' culture and daily life, in addition to standardized examinations. However, these existing benchmarks are designed in Simplified Chinese, thus, unsuitable for evaluating LLMs developed in the context of Traditional Chinese.

To the best of our knowledge, only two benchmarks—TC-Eval (Hsu et al., 2023) and TMMLU-plus (Tam et al., 2024)—have been developed towards the goal of evaluating LLMs' capabilities for Traditional Chinese. TC-Eval consists of a collection of existing datasets, mainly from conventional NLP tasks, *e.g.* reading comprehension, summarization, and sentiment classification (Shao et al., 2018; STPI, 2020; Narayan et al., 2018; Maas et al., 2011), and a newly introduced, MMLU-like multiple-choice dataset, termed TMMLU. Recently, TMMLU-plus is proposed to focus on the assessment of advance knowledge in Taiwanese-Mandarin LLMs. Compared to TMMLU, TMMLU-plus is six times larger and contains questions spanning a broader range of subjects.

Our proposed TMLU differs from previous benchmarks in the following ways. Firstly, all questions within TMLU are originally in Traditional Chinese. On the other hand, a sizeable portion of TC-Eval is translated from English datasets (Narayan et al., 2018; Maas et al., 2011; Srivastava et al., 2023), which misaligns the goal of evaluating LLMs in the Taiwanese-Mandarin environment due to potential western geographical biases. For example, the XSum dataset (Narayan et al., 2018) adopted by TC-Eval is sourced from BBC articles, which are likely grounded in western economic and social context. Although TMMLU-plus is tailored specifically for Taiwanese Mandarin, we still observe instances in Simplified Chinese (Figure 6b).

Secondly, compared to TMLU, TMMLU-plus exhibits several potential issues that could compromise the reliability and robustness of the benchmark. In our preliminary investigation, we find that most questions in TMMLU-plus could be found on one single website.[2] In addition, we observe the presence of the website's content in mC4 (Xue et al., 2021), a widely adopted corpus for pre-training LMs. More quantitatively, we randomly sample 100 instances from TMMLU-plus and discover 91 of them are on the website. We also examine the full mC4 corpus and find 28,685 entries have URLs under the website. Overall, this

---

[2] https://yamol.tw/

|  | # Subjects | # Instance (%) |
|---|---|---|
| *group by discipline* | | |
| Social Science | 8 | 589 (19.76%) |
| STEM | 14 | 468 (15.70%) |
| Humanities | 9 | 1,009 (33.85%) |
| Taiwan Specific | 3 | 557 (18.69%) |
| Others | 3 | 358 (12.01%) |
| *group by level* | | |
| Middle School | 9 | 434 (14.56%) |
| High School | 17 | 875 (29.35%) |
| Professional | 11 | 1,672 (56.09%) |
| Total | 37 | 2,981 (100.00%) |

Table 2: Statistics of our proposed TMLU benchmark grouped by different categories of discipline and difficulty levels.

greatly exposes its robustness to test data contamination (Sainz et al., 2023b;a; Golchin & Surdeanu, 2023; Shi et al., 2023). To minimize such risks, we follow Huang et al. (2023) by collecting data sourced in PDF and Microsoft Word documents from the internet, instead of plain texts directly from the web. Moreover, TMMLU-plus contains unanswerable questions that require information (*e.g.*, images or tables) not accompanied in the dataset (Figure 6b).

Lastly, TMLU provides two distinct features not available in TC-Eval and TMMLU-plus: (1) manual-constructed few-shot CoT demonstrations to elicit reasoning; (2) standardized mathematical expressions (*e.g.*, symbols and equations) in LaTeX format for better clarity (Figure 6a). In sum, our proposed TMLU is better for the community to objectively evaluate the model capability due to its better transparency[3], localization, and robustness.

## 3 TMLU Benchmark

### 3.1 Overview

The TMLU benchmark is an evaluation suit tailored for assessing advanced knowledge and reasoning capability in LLMs under Taiwanese Mandarin, in the format of multiple-choice questions. TMLU contains a wide range of subjects spanning social science, STEM, humanities, Taiwan-specific content, and others, across middle school to professional levels. The goal of TMLU is to provide an evaluation suit, which is easy-to-use and accessible, for developers to gauge how their models would likely perform in the real-world Taiwanese Mandarin context.

### 3.2 Data Source

One of the main concerns of NLP evaluation in the age of LLMs is test data contamination (Sainz et al., 2023b;a; Golchin & Surdeanu, 2023; Shi et al., 2023), sometimes referred to as data leakage. That is, the models are (pre-)trained on data that belongs to the test split of a benchmark and subsequently evaluate on the same test split, making the evaluation results questionable. As these is generally no guarantee of building contamination-free evaluation benchmark, especially given how little information about the training data are disclosed by proprietary models, mitigating the risk of contamination is equally important.

To this end, we follow the design principle of Huang et al. 2023 and collect data that is sourced in the format of PDF or Microsoft Word documents, which are files downloaded from the website instead of readily available within website plain text. Specifically, we source our data from the following five websites from Taiwan:[4]

---

[3]The source of data collection is not disclosed in the TMMLU-plus paper.

[4]The sourced website links are provided in Table 4.

1. *Comprehensive Assessment Program for Junior High School Students* (CAP) – the standardized exam for Taiwanese 9-th grade students before they go to high schools or vocational schools.

2. *General Scholastic Ability Test* (GSAT) – the Taiwanese university entrance exam, focusing on fundamental knowledge and skills from the required materials in the first two years of students' high school studies.

3. *Advanced Subjects Test* (AST) – the other part of the Taiwanese university entrance exam, focusing on advanced knowledge of individual subjects to assess students' comprehension, reasoning, and analysis abilities. The scope covers all materials in the high school studies.

4. *Ministry of Examination Exams* (MOEX) – the exams from Ministry of Examination of Taiwan consists of two main categories, civil service exams and professional and technical exams, spanning a wide range of fields such as law, finance, medical, and pharmacy.

5. *Highway Bureau Questions Bank* (HB) – the theory test questions from the Highway Bureau of Taiwan, including questions such as car and motorcycle driving regulations for the driving license test.

## 3.3 Data Processing

All of our data is in the format of PDF or Microsoft Word document initially and processed into text format subsequently, by manual annotation with the aided of OCR toolkits.[5] Mathematical expressions are standardized into LaTeX format for STEM subjects that involve substantial symbols and equations, following prior works (Hendrycks et al., 2021; Huang et al., 2023). The produced LaTeX formulas are compiled and verified by annotators to ensure their correctness. In addition, all instances are validated by human for quality assurance to exclude unsuitable instances (*e.g.*, unanswerable questions that require accompanied images or tables which are not presented in the text body).

Our final resulting benchmark, TMLU, consists of a collection of 2,981 multiple-choice questions across 37 subjects, covering difficulty levels from middle school, high school, to professional. We categorize the subjects into five disciplines – social science, STEM, humanities, Taiwan specific, and other. Each subject is further divided into a development set consisting of five instances and a test set consisting of the remaining instances. Statistics of TMLU grouped by difficulty disciplines and levels are provided in Table 2.

## 3.4 Explanation Curation

To facilitate the evaluation of reasoning capability in Taiwanese-Mandarin LLMs, we manually construct Chain-of-Thought-like explanations for each development set instances. Specifically, the main source of our explanations (55.68%) are curated by leveraging the textbook website.[6] It provides corresponding textbook-level explanations for CAP, GSAT, and AST exam questions included in TMLU, in the format of PDF documents. Other sources for explanation curation include *(1)* question-and-answer websites or forum (26.49%); *(2)* GPT-4-preview (3.78%), where explanations are adopted only if the predicted answer is correct; *(3)* fully handcrafted (4.86%), which is composed by thoroughly searching the Web; *(4)* others (9.19%), where the exact source is either ambiguous or untraceable.

# 4 Experiments

We conduct evaluation on an array of advanced LLMs on TMLU, establishing a baseline for research and community reference. Following we describe the experimental details and results.

---

[5]https://mathpix.com/

[6]https://www.ehanlin.com.tw/app/index.html

以下選擇題為出自臺灣的考題，答案為其中一個選項。
Following are multiple-choice questions from exams in Taiwan. The answer is one of the choices.

問題：
某一性狀由體染色體上的一對等位基因所控制，A 為顯性，a 為隱性。今有一對 夫妻此性狀的基因型皆為 Aa，在不考慮突變的情況下，他們小孩的此種性狀可能會有幾種表現型？
Question:
A trait is controlled by a pair of alleles on the autosomes, with A being dominant and a being recessive. If both parents have the genotype Aa for this trait, without considering mutations, how many different phenotypes could their children potentially have for this trait?

(A) 3 (B) 2 (C) 4 (D) 1

正確答案：(B)
Answer: (B)

[ ⋯ other four demonstrations ⋯]

問題：
下列為四本書的書名，每本書的書名分別顯示出所要介紹的內容，書中會列舉一些植物詳細說明其特徵，則哪一本書最不可能以蘇鐵作為這些植物的主要例子？
Question:
Among the following are the titles of four books, each title indicates the content it intends to introduce, where some plants are listed with detailed descriptions of their characteristics. Which book is the least likely to use cycads as the main example of these plants?

(A)《種子的傳播》(B)《毬果構造解析》(C)《維管束植物簡介》(D)《花朵圖鑑》
(A) 《The Spread of Seeds》 (B) 《Analysis of the Structure of Conifer Cones》 (C) 《An Introduction to Vascular Plants》 (D) 《A Pictorial Guide to Flowers》

正確答案：(
Answer: (

Figure 2: An example prompt for few-shot direct answer evaluation on TMLU.

## 4.1 Experimental Setups

**Models.** We comprehensively benchmark 24 LLMs on TMLU. The adopted models are capable of processing Traditional-Chinese content and vary in size and developer organizations. Concretely, we conduct experiment on 6 closed-source, proprietary models via APIs, including models from OpenAI,[7] Anthropic, and Google. For open-weight models, we focus on models developed for Chinese, English, and multilingual purposes. This included Taiwan-LLM (Lin & Chen, 2023) and Breeze (Hsu et al., 2024) for Taiwanese Mandarin, as well as Yi (AI et al., 2024) and Qwen (Bai et al., 2023) for Simplified Chinese. For a complete list of the models evaluated, please refer to Table 3.

**Few-shot Evaluation.** The evaluation is conducted in few-shot with setups: direct answer and CoT. For each subject, the five instances from the development split are utilized as demonstrations (*i.e.*, five shots). The adoption of few-shot evaluation is consistent with benchmarks such as C-Eval (Huang et al., 2023) and MMLU (Hendrycks et al., 2020), and the convention in prior works (Touvron et al., 2023; Team et al., 2023; Achiam et al., 2023) for measuring LLM performance. The setting is believe to capture the inherent, underlying potential of LLMs more robustly and extrapolate to downstream task adaptation (Huang et al., 2023).

The evaluation prompt consists of the following components: the problem definition, a set of few-shot examples, and the input instance to query. For instruction-tuned (chat-oriented) models, we apply the accompanied chat templates if available. Compared with direct answer, CoT prompting includes an explanation (*i.e.*, rationale) triggered with "*Let's think*

---

[7] We adopt gpt-4-0125-preview and gpt-3.5-turbo-1106.

*step by step*", before the final answer prediction. Example prompts with TMLU instances are provided in Figure 2 (direct answer) and Figure 7 (CoT).

**Answer Extraction.** As show by Fourrier et al., 2023, different flavors of implementation in extracting predictions from models bring nontrivial impact to the evaluation results. To ensure a fair evaluation and address variations in model accessibility and prompting settings, we adopt two methods for answer extraction: (1) *Likelihood-based* method. The method aligns with the original implementation in MMLU (Hendrycks et al., 2021), which involves examining a set of candidate answer symbols (*i.e.*, {"A", "B", "C", "D"}) and selecting the symbol with the highest assigned token probability as the model's prediction. (2) *Generation-based* method. When direct determination of candidate token probabilities is impractical, we opt for the first token generated greedily that can be interpreted as an option code to derive the answer prediction.

For proprietary models, as access to the probability of each candidate answer symbol is restricted, we utilize the generation-based method for both direct and CoT prompting scenarios. In the case of open-source models, we employ the likelihood-based method for direct prompting. However, when employing CoT prompting, where the model must first generate an explanation for the queried instance, we are constrained to using the generation-based method. In instances of CoT prompting, we adjust the number of shots based on the model context length to avoid failure to produce the answer.

### 4.2   Results

The experimental results are presented in Table 3a and Table 3b for direct answer and CoT prompting, respectively. In general, flagship proprietary models demonstrate performance superior to open-weight counterparts, and models trained on Chinese data (*i.e.*, Simplified Chinese, Taiwanese Mandarin, and Multilingual models) exhibit substantial improvement over models focusing on English context. For direct answer prompting, the Breeze model – tailored for Taiwanese Mandarin with only 7 billions parameters – outperforms GPT-3.5. The top-performing open-weight model Yi, with 34 billions parameters, outperforms proprietary models such as Gemini-pro and Claude-instant, and is comparable to GPT-4.

In CoT prompting, proprietary models outperform the best open-source model by a significant margin of 7%∼27%. The results suggest an ample opportunity for enhancing reasoning capabilities via CoT prompting in open-source models. We provide more analysis between direct answer and CoT prompting in Section 5.

In sum, the established baseline suggests that TMLU offers a proper ground, with headroom for improvements, for evaluating LLMs in the context of Taiwanese Mandarin. Furthermore, the exhibited performance of Taiwanese-Mandarin LLMs still lack behind the Simplified-Chinese counterparts, substantiating the goal of TMLU to foster the development of localized, Taiwanese-Mandarin LLMs in the future.

## 5   Analysis

### 5.1   Robustness to Test Data Contamination

To further investigate the possibility of test data contamination mentioned in Section 2 and 3.2, we employ MIN-K% PROB (Shi et al., 2023), a reference-free method for detecting pre-training data from LLMs. Based on the hypothesis that an example seen by the model before (*i.e.*, during pre-training) is less likely to include words with low probability (*i.e.*, high negative log-likelihood), MIN-K% PROB selects a set of $k\%$ of tokens from the input text with minimum token probability and average their log-likelihood as an indicator of whether the input text is in the pre-training data. Specifically, given an input example $x = (x_1, x_2, ..., x_n)$,

$$\text{MIN-K\% PROB}(x) = \frac{1}{|\text{Min-K}(x)|} \sum_{x_i \in \text{Min-K}(x)} -\log p(x_i | x_1, x_2, ..x_{i-1})$$

| Model | Social Sci. | STEM | Humanities | Taiwan Spe. | Other | Average |
|---|---|---|---|---|---|---|
| Claude-3-Opus$^\pi$ | **83.06** | **64.32** | 75.93 | **90.41** | 54.23 | **73.59** |
| Llama-3-Taiwan-70B-Instruct | 82.61 | 61.17 | **78.87** | 85.01 | 54.64 | 72.46 |
| GPT-4-turbo$^\pi$ | 78.14 | 58.79 | 72.30 | 86.90 | 55.98 | 70.42 |
| Yi-34B-Chat | 73.77 | 47.99 | 73.55 | 83.95 | **62.97** | 68.45 |
| Llama-3-70B-Instruct | 74.39 | 56.41 | 74.60 | 79.77 | 52.04 | 66.26 |
| Gemma-2-27b-it | 68.84 | 50.95 | 64.04 | 76.72 | 52.52 | 62.61 |
| Qwen-14B-Chat | 64.85 | 45.98 | 64.83 | 79.15 | 54.23 | 61.81 |
| Gemini-Pro$^\pi$ | 69.22 | 41.96 | 65.87 | 81.55 | 48.40 | 61.40 |
| Llama-3-Taiwan-8B-Instruct | 59.89 | 39.21 | 64.85 | 69.75 | 43.62 | 55.46 |
| Qwen-7B-Chat | 57.74 | 39.20 | 56.54 | 73.99 | 46.65 | 54.82 |
| Claude-Instant-1.2$^\pi$ | 60.84 | 36.43 | 56.33 | 75.46 | 42.86 | 54.38 |
| Breeze-7B-Instruct | 57.19 | 36.68 | 49.79 | 73.80 | 39.07 | 51.31 |
| GPT-3.5$^\pi$ | 54.64 | 35.43 | 46.37 | 72.51 | 37.90 | 49.37 |
| Mixtral-8x7B-Instruct | 51.55 | 33.17 | 46.47 | 66.05 | 34.99 | 46.44 |
| chatglm3-6b | 51.55 | 33.17 | 46.47 | 66.05 | 34.99 | 46.44 |
| Claude-2.0$^\pi$ | 40.62 | 30.90 | 41.08 | 73.06 | 37.32 | 44.60 |
| Mistral-7B-Instruct | 46.08 | 30.65 | 42.43 | 67.71 | 34.40 | 44.26 |
| Taiwan-LLM-13B-Chat | 44.08 | 27.89 | 41.80 | 64.39 | 36.15 | 42.86 |
| Baichuan2-13B-Chat | 46.08 | 31.91 | 42.84 | 56.64 | 28.57 | 41.21 |
| Taiwan-LLM-7B-Chat | 35.15 | 21.86 | 32.99 | 64.02 | 32.94 | 37.39 |
| falcon-40b-instruct | 36.07 | 21.36 | 36.41 | 51.48 | 32.07 | 35.48 |
| OLMo-7B-Instruct | 35.70 | 25.38 | 29.88 | 50.37 | 31.78 | 34.62 |
| Llama-2-13b-Chat | 36.07 | 27.14 | 35.27 | 47.05 | 26.24 | 34.35 |
| Yi-6B-Chat | 35.52 | 26.38 | 37.66 | 34.50 | 29.74 | 32.76 |
| Qwen-0.5B-Chat | 30.78 | 24.12 | 29.15 | 45.57 | 28.57 | 31.64 |
| Gemma-2-9b-it | 21.93 | 18.21 | 29.74 | 48.68 | 24.06 | 28.52 |
| Llama-2-7b-chat | 29.33 | 23.37 | 27.28 | 32.84 | 27.41 | 28.04 |
| TAIDE-LX-7B-Chat | 24.41 | 25.13 | 23.86 | 33.21 | 22.16 | 25.75 |
| Falcon-7b-instruct | 22.59 | 22.86 | 23.76 | 34.32 | 24.78 | 25.66 |
| Gemma-7b-it | 27.32 | 20.10 | 24.07 | 32.66 | 23.91 | 25.61 |
| Meta-Llama-3-8B | 17.75 | 16.41 | 19.81 | 22.66 | 24.33 | 20.19 |

(a) The five-shot direct-answer accuracy results. (%)

| Model | Social Sci. | STEM | Humanities | Taiwan Spe. | Other | Average |
|---|---|---|---|---|---|---|
| Claude-3-Opus$^\pi$ | **82.15** | **74.37** | **73.44** | **89.67** | **53.06** | **74.54** |
| GPT-4-turbo$^\pi$ | 75.77 | 74.12 | 70.12 | 86.35 | 51.31 | 71.54 |
| Gemini-Pro$^\pi$ | 64.66 | 47.24 | 59.54 | 82.66 | 45.77 | 59.97 |
| Claude-2.0$^\pi$ | 66.67 | 54.02 | 58.51 | 80.81 | 39.65 | 59.93 |
| Claude-Instant-1.2$^\pi$ | 61.57 | 44.47 | 56.43 | 75.83 | 37.03 | 55.07 |
| GPT-3.5$^\pi$ | 59.20 | 45.48 | 49.90 | 75.65 | 41.69 | 54.38 |
| Llama-3-Taiwan-70B-Instruct | 58.14 | 40.67 | 38.00 | 81.38 | 41.20 | 51.88 |
| Yi-6B-Chat | 46.45 | 34.17 | 46.27 | 70.85 | 39.07 | 47.36 |
| Taiwan-Llama-3-8B-Instruct | 45.72 | 29.15 | 41.49 | 66.97 | 31.20 | 42.91 |
| Yi-34B-Chat | 49.54 | 18.84 | 30.39 | 79.34 | 33.53 | 42.33 |
| Breeze-7B-Instruct | 48.63 | 22.61 | 30.19 | 74.35 | 27.11 | 40.58 |
| Mixtral-7B-Instruct | 42.81 | 23.87 | 37.14 | 59.23 | 30.61 | 38.73 |
| Llama-3-Taiwan-8B-Instruct | 38.62 | 25.56 | 38.00 | 60.89 | 26.88 | 37.99 |
| TAIDE-LX-7B-Chat | 38.25 | 26.38 | 32.05 | 67.34 | 24.78 | 37.76 |
| Gemma-2-9b-it | 37.56 | 32.91 | 34.95 | 50.01 | 30.20 | 37.13 |
| Qwen-7B-Chat | 39.71 | 18.09 | 29.25 | 64.94 | 30.03 | 36.41 |
| chatglm3-6b | 37.34 | 21.11 | 32.47 | 63.10 | 23.91 | 35.58 |
| Llama-3-70B-Instruct | 40.43 | 32.48 | 24.08 | 42.38 | 27.36 | 32.54 |
| Taiwan-LLM-7B-Chat | 26.05 | 23.62 | 19.81 | 56.27 | 24.49 | 30.05 |
| Gemma-2-27b-it | 29.26 | 23.67 | 27.04 | 39.48 | 18.26 | 27.54 |
| Baichuan2-13B-Chat | 29.51 | 19.85 | 27.07 | 30.26 | 27.11 | 26.76 |
| Taiwan-LLM-13B-Chat | 26.41 | 12.56 | 17.63 | 48.15 | 14.29 | 23.81 |
| Meta-Llama-3-8B | 25.04 | 20.23 | 23.62 | 23.48 | 23.51 | 23.18 |

(b) The five-shot CoT accuracy results. (%)

Table 3: The five-shot accuracy (%) results on TMLU for direct-answer and CoT prompting. The results are sorted in descending order. The colors indicate the language for which the models are tailored. Yellow denotes Multilingual; Blue denotes Simplified Chinese; Red denotes Taiwanese Mandarin; Green denotes English. In addition, proprietary models are marked with $\pi$. Detailed comparison results are provided at Table 9.

where Min-K(x) is the set of $k$% of tokens in $x$ with the minimum token probability. The lower the MIN-K% PROB is, the more likely that the input example $x$ has been seen during pre-training.

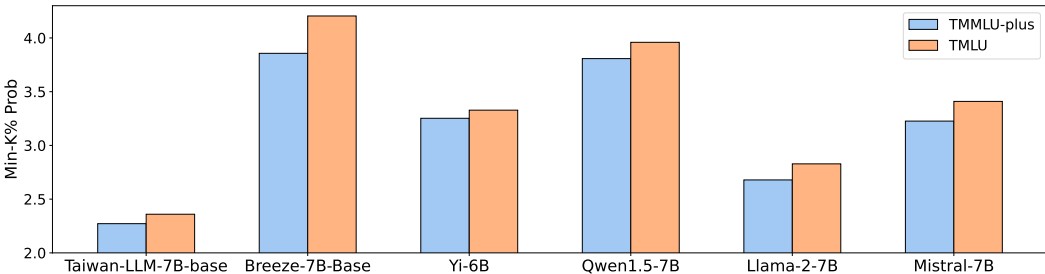

Figure 3: The MIN-K% PROB (Shi et al., 2023) of six base models on TMMLU-plus (Tam et al., 2024) and our dataset TMLU. The lower the MIN-K% PROB is, the more likely the input instances of the datasets are in the model's pre-training data.

We apply MIN-K% PROB on 2000 sampled instances from TMLU and TMMLU-plus with 6 different base models. We present results in Table 3 and discuss more implementation details in Appendix B. As shown, TMLU achieves higher MIN-K% PROB than TMMLU-plus across all tested models, which implies that TMLU is less likely to contain data that has been seen during pre-trained. In addition, the finding also aligns with our hypothesis that TMMLU-plus might be collected by crawling plain text from the web, and further validates the effectiveness of our design principle (Section 3.2) that sourcing data from downloaded documents could reduce the risk of data contamination.

## 5.2 Comparison Between Direct Answer and CoT Prompting

To further investigate the potential capability of LLMs, we leverage our curated explanations and employ CoT prompting. CoT prompting elicits step-by-step reasoning chains towards answer derivations, which has been shown to improve tasks that require complex, multi-hop reasoning significantly. We present results comparing CoT with standard answer-only prompting in Figure 4 for STEM subjects. Full results are reported in Table 9. As shown, models that are able to benefit from CoT are mainly large, proprietary ones, exemplified by GPT-4 (15%) and Claude-2 (23%). The finding may be attribute to the emergent nature of reasoning capability in LLMs. Interestingly, Taiwan-LLM-7B-Chat is the only model exhibiting improvement with CoT at its parametric scale.

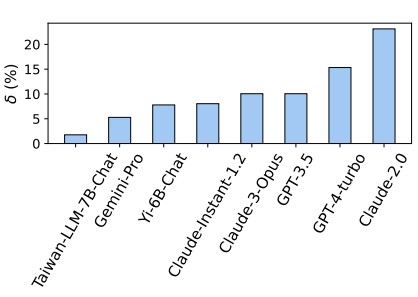

Figure 4: Performance difference ($\delta$) between direct answer and CoT prompting on stem subjects. Only models exhibiting improvements ($\delta > 0$) are presented.

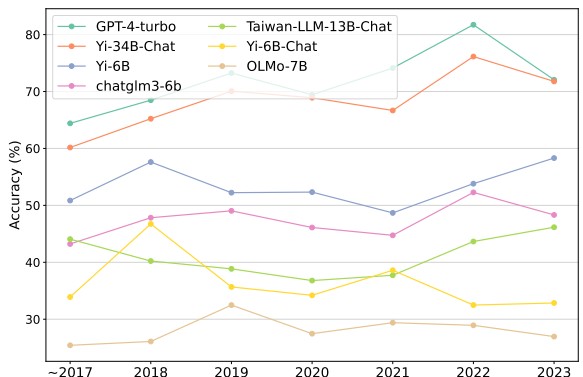

Figure 5: Average accuracy of models on questions of different years. The accuracy is calculated by averaging across the number of questions. Full results are provided at Table 6.

## 5.3 Comparison of Model Performance Across the Temporal Dimension

Here we investigate the performance of models on the temporal dimension. We group questions from different years as subsets and evaluate the average accuracy, as presented in Figure 5. Questions from before 2017 (2013∼2016) are group with 2017 as one subset. The results show that, for most models, no consistent trends or behaviors are exhibited. Yet, for GPT-4-turbo and Yi-34B-Chat, the scores imply a rising trend where models perform better on questions from more recent years. Future endeavors could offer a more precise implication and rigorous explanation for the results.

## 6 Conclusion

In this work, we introduce TMLU – an evaluation suite designed to assess the advanced knowledge and reasoning abilities of LLMs in the context of Taiwanese Mandarin. Experimental results and analysis suggest great opportunities for future developers, in particular, for open-source models. Besides addressing the scarcity of benchmark for Traditional Chinese community, we envision TMLU to serve as a grounded testbad, fostering development of localized Taiwanese-Mandarin LLMs.

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

## A  Limitations and Future Work

While TMLU provides a comprehensive evaluation of language models' understanding capabilities in Taiwanese Mandarin, it is important to acknowledge its limitations, particularly in terms of its generalizability to generation tasks. Our benchmark, which focuses on multiple-choice question answering, may not fully capture the performance of language models in generation scenarios such as text completion, summarization, and dialogue systems.

To investigate this issue, we conduct a preliminary experiment by calculating the perplexity (PPL) of generated answers for a few models with a fixed vocabulary size of 32,000 tokens. The resulting PPL values were: Taiwan-LLM-7B-Chat: 16.5, Breeze-7B-instruct: 8836.1, Yi-6B-Chat: 20887.0, and Llama-7B-Chat: 2038633.8. Interestingly, models trained exclusively on Taiwanese Mandarin, such as Taiwan-LLM, have a smaller entropy in their logits compared to models trained on multilingual data, suggesting more consistent performance in generating fluent Taiwanese Mandarin text.

Therefore, while TMLU provides valuable insights into the understanding capabilities of language models in Taiwanese Mandarin, it should not be considered a sole indicator of their performance in generation tasks. Future research should focus on developing generation-oriented benchmarks and exploring the relationship between understanding and generation

| Data source | Website |
| --- | --- |
| CAP | https://cap.rcpet.edu.tw/examination.html |
| GSAT | https://www.ceec.edu.tw/xmfile?xsmsid=0J052424829869345634 |
| AST | https://www.ceec.edu.tw/xmfile?xsmsid=0J052427633128416650 |
| MOEX | https://wwwq.moex.gov.tw/exam/wFrmExamQandASearch.aspx |
| HB | https://www.thb.gov.tw/News_Download.aspx?n=284&sms=12823 |

Table 4: The sourced websites of our data collection.

| Supercategory | Subjects | # Questions | # Choices |
| --- | --- | --- | --- |
| Social Science | AST civics (分科測驗公民) | 57 | 4 |
| | AST geography (分科測驗地理) | 58 | 4 |
| | CAP civics (會考公民) | 73 | 4 |
| | CAP geography (會考地理) | 45 | 4 |
| | GSAT civics (學測公民) | 73 | 4 |
| | GSAT geography (學測地理) | 49 | 4 |
| | MOEX Accountant (會計師) | 117 | 4 |
| | MOEX Clinical psychologist (臨床心理師) | 117 | 4 |
| STEM | AST biology (分科測驗生物) | 40 | 4 |
| | AST chemistry (分科測驗化學) | 34 | 5 |
| | AST mathematics (分科測驗數學) | 25 | 5 |
| | AST physics (分科測驗物理) | 43 | 5 |
| | CAP biology (會考生物) | 27 | 4 |
| | CAP chemistry (會考化學) | 27 | 4 |
| | CAP earth science (會考地球科學) | 15 | 4 |
| | CAP mathematics (會考數學) | 115 | 4 |
| | CAP physics (會考物理) | 15 | 4 |
| | GSAT biology (學測生物) | 21 | 5 |
| | GSAT chemistry (學測化學) | 29 | 5 |
| | GSAT earth science (學測地球科學) | 24 | 5 |
| | GSAT mathematics (學測數學) | 29 | 5 |
| | GSAT physics (學測物理) | 24 | 5 |
| Humanities | AST Chinese (分科測驗國文) | 131 | 4 |
| | AST history (分科測驗歷史) | 56 | 4 |
| | CAP Chinese (會考國文) | 61 | 4 |
| | CAP history (會考歷史) | 56 | 4 |
| | GSAT Chinese (學測國文) | 97 | 4 |
| | GSAT history (學測歷史) | 85 | 4 |
| | MOEX Tour guide (導遊) | 99 | 4 |
| | MOEX Tour leader (領隊) | 145 | 4 |
| | MOEX Lawyer qualification (律師資格) | 279 | 4 |
| Taiwan Specific | HB Driving Rule (台灣駕駛規則) | 432 | 4 |
| | MOEX Teacher qualification (教師資格) | 75 | 4 |
| | MOEX Taiwan tourist resources (台灣觀光資源) | 50 | 4 |
| Others | MOEX Basic Traditional Chinese Medicine (中醫基礎醫學) | 159 | 4 |
| | MOEX Clinical Traditional Chinese Medicine (中醫針灸) | 79 | 4 |
| | MOEX Nutritionist (營養師) | 120 | 4 |

Table 5: Summary of all 37 subjects and corresponding numbers of choices.

abilities to gain a more holistic view of the performance of language models in the context of Taiwanese Mandarin.

## B  Implementation Details of Data Contamination Analysis

We construct the input example $x$ for MIN-K% PROB by concatenating the question and the corresponding choices – consistent to the actual scenario where the model would likely be queried – for each sampled instances from TMLU and TMMLU-plus. To mitigate the effect of input example length when computing MIN-K% PROB, as discussed in the MIN-K% PROB paper, we filtered out the longest examples within each subjet subset of our dataset, as the average text length of TMLU is significantly longer than that of TMMLU-plus. This step is taken to ensure a fair and comparable analysis.

[TMLU] - The instance from "AST physics" test split.

已知氫原子的能階公式為 $E_{n}=-13.6 / n^{2} \mathrm{eV}$ ，其中 $n$ 為主量子數。一個動能為 $12.3 \mathrm{eV}$ 的電子與基態的氫原子發生碰撞, 下列何者可能為激發後的氫原子所發出的光子能量？
It is known that the energy level formula for a hydrogen atom is $E_{n}=-13.6 / n^{2} \mathrm{eV}$, where $n$ is the principal quantum number. An electron with a kinetic energy of $12.3 \mathrm{eV}$ collides with a hydrogen atom in its ground state. Which of the following could be the energy of a photon emitted by the hydrogen atom after excitation?

[TMMLU-plus] - The instance from "physics" test split.

已知氫原子的能階公式為 $En=-13.6/n2eV$，其中 $n$ 為主量子數。一個動能為 $12.3eV$ 的電子與基態的氫原子發生碰撞，下列何者可能為激發後的氫原子所發出的光子能量？
It is known that the energy level formula for a hydrogen atom is $En = -13.6/n^2$ eV, where $n$ is the principal quantum number. An electron with a kinetic energy of $12.3$ eV collides with a hydrogen atom in its ground state. Which of the following could be the energy of the photon emitted by the hydrogen atom after excitation?

(a)

[TMMLU-plus] - A Simplified-Chinese instance from the "physics" test split.

通过提高光的以下哪项可以使光的强度变高？
Which of the following factors, if increased, can cause the intensity of light to increase?

[TMMLU-plus] - Two unanswerable instances from the "junior social studies" test split.

根據圖(十六)判斷，關於這則宣傳標語所呈現的意義，下列敘述何者最適當？
Based on Figure 16, which of the following descriptions most appropriately represents the meaning conveyed by this promotional slogan?

根據表(九)中資訊，小婷與體力、經驗較為不足的民眾，選擇的路線分別為下列何者？
According to the information in Table 9, which route did Xiaoting and the people with less physical strength and experience choose?

(b)

Figure 6: Example instances from TMLU and TMMLU-plus datastes. Here choices are omitted for brevity. Figure 6a shows a same instance from TMLU (AST_physics) and TMMLU-plus (physics_test), where symbols are formatted in latex in TMLU but are not formatted in TMMLU-plus, effecting the clarity. Figure 6b shows instances from TMMLU-plus (junior_chemistry_test and junior_social_studies_test) which are in Simplified Chinese or unanswerable since it requires information from an image/table that is not accompanied in the question.

以下選擇題為出自臺灣的考題，答案為其中一個選項。
Following are multiple-choice questions from exams in Taiwan. The answer is one of the choices.

問題：
下列有關植物組織的敘述何者正確?
Question:
Which of the following statements about plant tissues is correct?

(A) 具有光合作用的葉肉細胞是屬於厚壁細胞
(B) 氣孔是由細胞間隙形成
(C) 生組織的細胞壁比葉肉細胞厚
(D) 木栓形成層由表皮細胞分化而成
(A) The mesophyll cells, which perform photosynthesis, are a type of sclerenchyma cells.
(B) Stomata are formed by intercellular spaces.
(C) The cell walls of meristematic tissue are thicker than those of mesophyll cells.
(D) The cork cambium is differentiated from epidermal cells.

讓我們一步一步思考。具有光合作用的葉肉細胞是屬於薄壁細胞，故(A)錯誤。(B)敘述正確，應選(B)。分生組織的細胞壁比葉肉細胞厚薄，故(C)錯誤。形成層以外的細胞都可轉為木栓形成層，但不包括表皮細胞，故(D)錯誤。
正確答案：(B)
Let's think step by step. The mesophyll cells with photosynthesis belong to parenchyma cells, so (A) is incorrect. (B) The description is correct, so choose (B). The cell walls of the meristematic tissue are thicker than those of the mesophyll cells, so (C) is incorrect. Cells outside of the cambium can all turn into cork cambium, except for the epidermal cells, so (D) is incorrect.
Answer: (B)

[ ⋯ other four demonstrations ⋯ ]

問題：
下列遺傳篩檢技術中, 何者最能確診胎兒罹患唐氏症?
Question:
Among the following genetic screening technologies, which is most capable of diagnosing fetal Down syndrome?

(A) 超音波檢驗 (B) 染色體檢驗 (C) 基因檢驗 (D) 生化檢驗
(A) Ultrasound examination (B) Chromosomal examination (C) Genetic testing (C) Biochemical testing

讓我們一步一步思考。
Let's think step by step.

Figure 7: An example prompt for few-shot CoT evaluation on TMLU.

| Model | ~2017 | 2018 | 2019 | 2020 | 2021 | 2022 | 2023 | Avg. |
|---|---|---|---|---|---|---|---|---|
| # of questions | 118 | 92 | 157 | 193 | 228 | 197 | 1811 | |
| GPT-4-turbo | 64.41 | 68.48 | 73.25 | 69.43 | 74.12 | 81.73 | 72.06 | 72.35 |
| Yi-34B-Chat | 60.17 | 65.22 | 70.06 | 68.91 | 66.67 | 76.14 | 71.78 | 70.67 |
| Gemini-pro | 56.78 | 65.22 | 63.06 | 61.14 | 61.40 | 67.51 | 64.77 | 64.02 |
| Qwen-7B-Chat | 48.31 | 56.52 | 59.87 | 51.30 | 54.39 | 57.36 | 57.43 | 56.47 |
| Claude-instant-1.2 | 49.15 | 56.52 | 55.41 | 48.70 | 52.19 | 56.85 | 58.31 | 56.44 |
| Yi-6B | 50.85 | 57.61 | 52.23 | 52.33 | 48.68 | 53.81 | 58.31 | 56.12 |
| GPT-3.5 | 51.69 | 44.57 | 52.23 | 44.04 | 41.67 | 55.33 | 51.79 | 50.46 |
| Mixtral-8x7B-Instruct | 43.22 | 51.09 | 52.23 | 43.01 | 42.98 | 54.31 | 50.19 | 49.25 |
| chatglm3-6b | 43.22 | 47.83 | 49.04 | 46.11 | 44.74 | 52.28 | 48.32 | 47.96 |
| Mistral-7B-Instruct | 39.83 | 47.83 | 43.31 | 42.49 | 28.95 | 38.58 | 48.92 | 45.39 |
| Claude-2.0 | 38.98 | 46.74 | 40.76 | 38.34 | 41.23 | 42.64 | 47.54 | 45.28 |
| Taiwan-LLM-13B-Chat | 44.07 | 40.22 | 38.85 | 36.79 | 37.72 | 43.65 | 46.16 | 43.96 |
| Baichuan2-13B-Chat | 37.29 | 42.39 | 50.96 | 38.34 | 40.35 | 54.31 | 42.08 | 42.85 |
| Taiwan-LLM-7B-Chat | 27.97 | 29.35 | 32.48 | 30.57 | 27.63 | 33.50 | 41.91 | 37.84 |
| falcon-40b-instruct | 30.51 | 33.70 | 34.39 | 35.23 | 31.58 | 36.55 | 38.10 | 36.59 |
| Llama-2-13b-Chat | 40.68 | 34.78 | 32.48 | 29.53 | 32.89 | 40.61 | 35.78 | 35.44 |
| Yi-6B-Chat | 33.90 | 46.74 | 35.67 | 34.20 | 38.60 | 32.49 | 32.85 | 34.05 |
| Qwen-0.5B-Chat | 35.59 | 27.17 | 26.11 | 27.98 | 26.75 | 30.46 | 33.57 | 31.87 |
| Llama-2-7b-Chat | 32.20 | 32.61 | 28.03 | 22.80 | 30.26 | 24.87 | 28.44 | 28.22 |
| OLMo-7B | 25.42 | 26.09 | 32.48 | 27.46 | 29.39 | 28.93 | 26.95 | 27.54 |
| gemma-7b-it | 20.34 | 17.39 | 24.84 | 21.76 | 25.00 | 28.93 | 26.84 | 25.79 |
| falcon-7b-instruct | 14.41 | 20.65 | 23.57 | 20.21 | 26.75 | 30.96 | 26.56 | 25.57 |
| Total | 41.34 | 44.09 | 44.34 | 41.37 | 41.73 | 46.81 | 46.70 | 45.45 |

Table 6: Performance comparison on questions from different years. The accuracy is calculated by averaging across the number of questions.

| Year | # of questions |
|---|---|
| 2013 | 10 |
| 2014 | 15 |
| 2015 | 27 |
| 2016 | 27 |
| 2017 | 39 |
| 2018 | 92 |
| 2019 | 157 |
| 2020 | 193 |
| 2021 | 228 |
| 2022 | 197 |
| 2023 | 1811 |

Table 7: Number of questions from different years.

| Model | Social Science | | STEM | | Humanities | | Taiwan Specific | | Other | | Avg. | |
|---|---|---|---|---|---|---|---|---|---|---|---|---|
| | Base | Chat | Base | Chat | Base | Chat | Base | Chat | Base | Chat | Base | Chat |
| Yi-34B | 77.96 | 73.77 | 51.51 | 47.99 | 73.76 | 73.55 | 86.35 | 83.95 | 69.68 | 62.97 | 71.85 | 68.45 |
| Qwen-14B | 66.12 | 64.85 | 47.49 | 45.98 | 65.77 | 64.83 | 78.97 | 79.15 | 57.14 | 54.23 | 63.10 | 61.81 |
| Yi-6B | 60.29 | 35.52 | 40.20 | 26.38 | 60.27 | 37.66 | 78.04 | 34.50 | 50.44 | 29.74 | 57.85 | 32.76 |
| Qwen-7B | 60.47 | 57.74 | 39.45 | 39.20 | 60.37 | 56.54 | 75.46 | 73.99 | 52.19 | 46.65 | 57.59 | 54.82 |
| Baichuan2-13B | 55.56 | 46.08 | 31.41 | 31.91 | 53.63 | 42.84 | 72.69 | 56.64 | 46.36 | 28.57 | 51.93 | 41.21 |
| Mistral-7B | 51.91 | 46.08 | 31.41 | 30.65 | 43.46 | 42.43 | 68.08 | 67.71 | 35.28 | 34.40 | 46.03 | 44.26 |
| Taiwan-LLM-13B | 42.08 | 44.08 | 25.38 | 27.89 | 44.09 | 41.80 | 67.90 | 64.39 | 35.28 | 36.15 | 42.94 | 42.86 |
| Llama-2-13b | 44.44 | 36.07 | 31.66 | 27.14 | 39.32 | 35.27 | 62.36 | 47.05 | 32.36 | 26.24 | 42.03 | 34.35 |
| Qwen-0.5B | 36.61 | 30.78 | 26.88 | 24.12 | 35.58 | 29.15 | 52.58 | 45.57 | 26.82 | 28.57 | 35.70 | 31.64 |
| Taiwan-LLM-7B | 34.79 | 35.15 | 24.62 | 21.86 | 29.67 | 32.99 | 57.56 | 64.02 | 27.11 | 32.94 | 34.75 | 37.39 |
| Llama-2-7b | 31.69 | 29.33 | 26.88 | 23.37 | 29.98 | 27.28 | 55.54 | 32.84 | 23.32 | 27.41 | 33.48 | 28.04 |
| falcon-7b | 23.50 | 22.59 | 25.13 | 22.86 | 28.53 | 23.76 | 32.29 | 34.32 | 25.36 | 24.78 | 26.96 | 25.66 |
| OLMo-7B | 23.50 | 35.70 | 23.37 | 25.38 | 26.66 | 29.88 | 34.50 | 50.37 | 24.78 | 31.78 | 26.56 | 34.62 |
| gemma-7b | 25.14 | 27.32 | 23.62 | 20.10 | 24.79 | 24.07 | 29.34 | 32.66 | 25.07 | 23.91 | 25.59 | 25.61 |

Table 8: Performance comparison between base and instruction-tuned models.

| Model | Social Science | | STEM | | Humanities | | Taiwan Specific | | Other | | Avg. | |
|---|---|---|---|---|---|---|---|---|---|---|---|---|
| | Direct | CoT | Direct | CoT | Direct | CoT | Direct | CoT | Direct | CoT | Direct | CoT |
| Claude-3-Opus | 83.06 | 82.15 | 64.32 | 74.37 | 75.93 | 73.44 | 90.41 | 89.67 | 54.23 | 53.06 | 73.59 | 74.54 |
| GPT-4-turbo | 78.14 | 75.77 | 58.79 | 74.12 | 72.30 | 70.12 | 86.90 | 86.35 | 55.98 | 51.31 | 70.42 | 71.54 |
| Yi-34B-Chat | 73.77 | 49.54 | 47.99 | 18.84 | 73.55 | 30.39 | 83.95 | 79.34 | 62.97 | 33.53 | 68.45 | 42.33 |
| Qwen-14B-Chat | 64.85 | 55.19 | 45.98 | 25.63 | 64.83 | 45.85 | 79.15 | 78.04 | 54.23 | 48.98 | 61.81 | 50.74 |
| Gemini-Pro | 69.22 | 64.66 | 41.96 | 47.24 | 65.87 | 59.54 | 81.55 | 82.66 | 48.40 | 45.77 | 61.40 | 59.97 |
| Qwen-7B-Chat | 57.74 | 39.71 | 39.20 | 18.09 | 56.54 | 29.25 | 73.99 | 64.94 | 46.65 | 30.03 | 54.82 | 36.41 |
| Claude-Instant-1.2 | 60.84 | 61.57 | 36.43 | 44.47 | 56.33 | 56.43 | 75.46 | 75.83 | 42.86 | 37.03 | 54.38 | 55.07 |
| Breeze-7B-Instruct | 57.19 | 48.63 | 36.68 | 22.61 | 49.79 | 30.19 | 73.80 | 74.35 | 39.07 | 27.11 | 51.31 | 40.58 |
| GPT-3.5 | 54.64 | 59.20 | 35.43 | 45.48 | 46.37 | 49.90 | 72.51 | 75.65 | 37.90 | 41.69 | 49.37 | 54.38 |
| ChatGLM3-6B | 51.55 | 37.34 | 33.17 | 21.11 | 46.47 | 32.47 | 66.05 | 63.10 | 34.99 | 23.91 | 46.44 | 35.58 |
| Claude-2.0 | 40.62 | 66.67 | 30.90 | 54.02 | 41.08 | 58.51 | 73.06 | 80.81 | 37.32 | 39.65 | 44.60 | 59.93 |
| Mistral-7B-Instruct | 46.08 | 42.81 | 30.65 | 23.87 | 42.43 | 37.14 | 67.71 | 59.23 | 34.40 | 30.61 | 44.26 | 38.73 |
| Taiwan-LLM-13B-Chat | 44.08 | 26.41 | 27.89 | 12.56 | 41.80 | 17.63 | 64.39 | 48.15 | 36.15 | 14.29 | 42.86 | 23.81 |
| Baichuan2-13B-Chat | 46.08 | 29.51 | 31.91 | 19.85 | 42.84 | 27.07 | 56.64 | 30.26 | 28.57 | 27.11 | 41.21 | 26.76 |
| Taiwan-LLM-7B-Chat | 35.15 | 26.05 | 21.86 | 23.62 | 32.99 | 19.81 | 64.02 | 56.27 | 32.94 | 24.49 | 37.39 | 30.05 |
| Llama-2-13v-Chat | 36.07 | 23.13 | 27.14 | 20.10 | 35.27 | 28.53 | 47.05 | 39.30 | 26.24 | 26.24 | 34.35 | 27.46 |
| Yi-6B-Chat | 35.52 | 46.45 | 26.38 | 34.17 | 37.66 | 46.27 | 34.50 | 70.85 | 29.74 | 39.07 | 32.76 | 47.36 |
| Qwen-0.5B-Chat | 30.78 | 22.04 | 24.12 | 18.59 | 29.15 | 20.02 | 45.57 | 34.69 | 28.57 | 18.08 | 31.64 | 22.68 |
| Llama-2-7B-Chat | 29.33 | 9.47 | 23.37 | 7.04 | 27.28 | 7.68 | 32.84 | 1.66 | 27.41 | 9.33 | 28.04 | 7.03 |

Table 9: Performance comparison between direct and CoT prompting.

