# OpenReview forum: "Measuring Taiwanese Mandarin Language Understanding"
_colmweb.org/COLM/2024/Conference — COLM_

### Official Review · Reviewer_u2Ve · 2024-05-05

**Rating:** 7
**Confidence:** 3
**Ethics Flag:** 1

**Summary:**

The paper presents a new evaluation dataset (TMLU) for traditional Mandarin Chinese (Taiwanese).
It contains 37 subjects (social science, STEM, humanities and Taiwan specific). The contributions of this eval dataset and this paper are:

It aims at better evaluating LLMs on traditional Chinese, which has unique characteristics and challenges compared to simplified Chinese
Compared to existing traditional Chinese evaluation datasets (TC-eval and TMMLU-plus), TMLU has the following advantages:
1. not from a single website, using pdfs or words, not easily scrapable so avoiding **data contaminations**;
2. more answerable from the question context;
3. including CoT instructions and Latex expressions

**Questions To Authors:**

1. From my understanding “同志” can mean “comrade” and “homosexual” depending on the context even in simplified Chinese. I still agree with the point that words carry different meanings in simplified and traditional Chinese, and the challenge do exist for LLM evaluations, but this may not be the best example to showcase in the Introduction section?
2. If I understand correctly, the data comes exclusively from examinations (high school, college or professional), that may impose some bias into this dataset in a way that we are encouraging LLMs to be good at answering those “non-daily” questions, but does not necessarily make models more “usable” at answering “daily” questions. So it mainly measures knowledge, not necessarily other aspects of language understanding, such as understanding the nuances in languages and LLMs’ usability and “chattability”. Think about people’s daily usage of LLMs, they may ask questions about trip planning, shopping ideas, financial knowledge, legal advice, emotional support, relationship questions, career advice, business knowledge, etc. I would like to see those covered as well. This could be out of the scope of this paper, though. I hope we will have more such kind of evaluation data in the future.
3. Figure 3 shows that TMLU is less likely to be included in training data of these benchmarked LLMs, but not by a big margin. What are your takeaways on these results?

**Reasons To Accept:**

1. It is great to see a new dataset introduced to address the evaluation needs of traditional Chinese (Taiwan) languages.
2. Authors demonstrate TMLU has advantages as an evaluation dataset compared to existing traditional Chinese eval benchmarks, including better handling of data contamination, less unanswerable questions.

**Reasons To Reject:**

Although authors show the dataset has advantages compared to existing benchmarks, I hope the dataset can do even better. In particular, I hope the eval dataset can cover the following capabilities:
1. Chatability: being able to understand a dialogue context
2. Understanding nuances from languages
3. Other use cases in addition to exams

Please see my questions below.

---

> ### Author Rebuttal · Authors · 2024-05-31
>
> Thanks for the positive comment!
>
> **Response to Q1**
>
> Thank you for pointing this out! We apologize for not fully comprehending all the meaning of the term "同志" in Simplified Chinese. We will revise the illustrative example with other terms such as "土豆" (which typically means “potato” in Mainland China and “peanut” in Taiwan).
>
> **Response to Q2**
>
> We acknowledge your concern regarding the limited scope of our dataset, which currently only evaluates knowledge without assessing other communicative skills. Although the primary focus of our dataset is to evaluate the LLM's proficiency in Taiwanese-Mandarin knowledge, we are committed to augment TMLU in the future to include more diverse data related to “non-daily” tasks and interactive conversation in Taiwanese Mandarin. Thanks for your constructive suggestion!
>
> **Response to Q3**
>
> Thanks for the insightful question! The metric adopted in Figure 3, Min-K%, is an indicator of how likely the given text is in the pre-training data of the examined LM. The absolute number of Min-K% varies under different settings (e.g., model size, vocabulary size), and requires oracle labels of the tested dataset to determine a threshold that reflects whether contamination occurs (Please refer to Section 5.1 in the Min-K% paper).
>
> Nonetheless, we believe a systematic and unifying behavior in Figure 3 (where TMMLU-plus records a worse number in every model tested) could imply that TMLU is less likely to be contaminated.

---

### Official Review · Reviewer_L2nz · 2024-05-10

**Rating:** 7
**Confidence:** 5
**Ethics Flag:** 1

**Summary:**

This paper describes TMLU, a new MMLU-style dataset for LLM evaluation, targeted specifically at Taiwanese Mandarin (and the traditional Chinese script). As with MMLU and other datasets fashioned around it, the source of the data for TMLU is exam questions from middle and high school, and professional topics (e.g. traditional Chinese medicine or driving exams), with a strong focus on scraping data from PDFs and MSWord documents in the hope that it is less likely to be included in the pre-training data for LLMs. In addition to evaluating a range of open-weight and proprietary LLMs over TMLU, the paper includes some nice analysis of data contamination, direct answering vs. COT prompting, and a breakdown of results across time (based on the timestamps of the documents the questions were sourced from).

**Questions To Authors:**

- overall, models appear to achieve slightly lower results over TMLU than CMMLU, e.g., but how much of this is attributable to script differences (traditional vs. simplified) vs. the Taiwanese/Mandarin Chinese distinction vs. "local knowledge"? For example, if you convert TMLU into the simplified script, how much do the results change? How do you explain the fact that results over "Taiwan specific" questions are actually higher than all other question types?
- in what sense is TMLU "holistic" (page 8)? I get that it is broad coverage, but is there a way to quantify the degree of comprehensiveness of the dataset (e.g. relative to MMLU or CMMLU)?
- more of a comment than a question, but I found myself getting confused at one point about what TMLU was as compared to TMMLU-plus (i.e. given the similarity between "TMLU" and "TMMLU", I assumed that TMMLU-plus was somehow an extended version of TMLU and tried to go back and find mention of how it had been expanded); perhaps remind the reader in Section 5.1 that they are completely different datasets, despite the similarity in name
- what is the basis for you saying that "the applicability of these benchmarks has decreased as the ever larger models are demonstrating human-level performance"? yes, models are improving, but there is a bit of circularity in the slightly awkward wording here, as "human level performance" is being claimed in large part due to improvement over benchmarks, but at the same time, data contamination is becoming a bigger and bigger issue; possibly nuance the claim a bit

**Reasons To Accept:**

- it is important for LLM evaluation resources to be developed for a broad range of languages, and while there are many resources for Mandarin Chinese, there are considerably fewer for Taiwanese Mandarin, and the traditional Chinese script; this combined with the scale of the dataset and attention to detail in the paper makes it worthy of publication
- beyond just presenting results for different LLMs over TMLU, the paper includes additional analysis of data contamination (showing that it is lower than a competitor dataset), direct answering vs. COT prompting, and LLM accuracy of questions sourced from different years

**Reasons To Reject:**

- not a strong reason to reject, but I would have liked to have seen more comparison of the coverage of topics in the dataset with related datasets such as TMMLU-plus and CMMLU
- I would also have liked to have seen more discussion/analysis of results over "Taiwan specific" questions (see below), and also between similar topics in CMMLU and other datasets for Mandarin Chinese
- the writing is overall good, but there were some places where things got tangled up, particularly the first sentence of Section 4.1 which should be changed  to "We perform comprehensive evaluation over TMLU using 24 different LLMs ..." (you are not evaluating TMLU itself, but rather using TMLU to evaluate LLMs); also "wide range of field*s*" (Section 3.2); "Tradition*al* Chinese (Section 2)

---

> ### Author Rebuttal · Authors · 2024-05-31
>
> Thanks for the positive feedback!
>
> **Response to RR1 & RR2**
>
> The topic coverage and composition structure of CMMLU, TMMLU-plus and TMLU are similar, generally divided into STEM, social sciences, humanities, local-specific, and others. CMMLU and TMMLU-plus includes more professional-level domain-specific subjects, such as computer science and medical knowledge. Compared to TMMLU-plus, TMLU has a more complete coverage of elementary and high school -level questions.
>
> In the local-specific section, CMMLU includes many subjects related to Chinese literature; TMLU includes subjects related to local tourism in Taiwan, such as tour guide exams and tourism resources, to evaluate the model’s Taiwan-specific knowledge, and TMMLU-plus does not have an explicit subset for localized questions.
>
> We will incorporate the above content and a more in-depth discussion/analysis in our revised version.
>
> **Response to RR3**
>
> Thanks for pointing this out! We will correct semantic errors and refine all necessary parts to improve clarity in our final version.
>
> **Response to Q1**
>
> We hypothesize Simplified Chinese LMs perform sufficiently well because they might have already been trained on extensive Traditional Mandarin contents (in addition to more Simplified Chinese corpus that contain Taiwan’s local knowledge). Also, a potential explanation is that the Taowan-specific questions are relatively easy compared to advanced STEM subjects, resulting in higher scores compared to other academic subjects. However, we can still observed in Table 3b – at the same parametric scale, Taiwanese-Mandarin model Breeze-7B-Instruct greatly outperms Simplified-Chinese model Qwen-7B-Chat (74.35% vs 64.94%) on the Taiwan-specific subset of TMLU.
>
> **Response to Q2**
>
> Initially we use the term “holistic” in the sense that our dataset covers a wide range of topics for assessing the models’ parametric knowledge. We will refine our description of “holistic” to be precise and try to provide a more quantified comparison of the degree of comprehensiveness.
>
> **Response to Q3 & Q4**
>
> Sorry for the confusion! Indeed TMMLU-plus is an extension of TMMLU, and both factors (data contamination issue and the improvement of model itself such as scaling) tangle together. We will reiterate the difference in Section 5.1 as suggested and nuance the description regarding the applicability of the benchmark in our final version.

---

### Official Review · Reviewer_8Aze · 2024-05-11

**Rating:** 5
**Confidence:** 4
**Ethics Flag:** 1

**Summary:**

This paper introduces TMLU, a new evaluation benchmark designed to assess large language models (LLMs) specifically in the context of Taiwanese Mandarin. TMLU addresses the unique linguistic nuances and cultural intricacies of Taiwanese Mandarin, differentiating it from other varieties of Chinese. The benchmark encompasses 37 subjects, spanning middle school to professional levels, and includes approximately 3,000 multiple-choice questions across diverse disciplines. The authors evaluate 24 LLMs on TMLU and include Chain-of-Thought (CoT) examples to assess complex reasoning skills.

**Questions To Authors:**

Figure 2 contains an error in the answer key (the correct answer for the last question is "C").
The example using "Tongzhi" to illustrate language differences is not the most effective, as its primary meaning is consistent across Mainland China and Taiwan (referencing Zhang et al. 2022).
Zhang, K., Lu, C., & Zhang, J. (2022). Chinese media representations of tongzhi (2009–2019). Discourse & Communication, 16(3), 305-325.

**Reasons To Accept:**

The paper introduces a novel benchmark, TMLU, designed specifically for evaluating the performance of large language models (LLMs) in Taiwanese Mandarin. This benchmark encompasses a broad spectrum of subjects and has been meticulously curated through manual review. Additionally, the authors attempt to mitigate data contamination by sourcing questions from PDF and Microsoft Word documents.

**Reasons To Reject:**

1. Limited Linguistic and Cultural Differentiation: While the benchmark is written in Traditional Chinese, the questions themselves do not sufficiently capture the unique linguistic and cultural aspects of Taiwanese Mandarin. Many STEM questions, for example, use terminology common to Mainland China. Areas where language usage differences are more pronounced, like social media, are not included. This limitation may explain why Simplified Chinese models perform well on the benchmark.
2. Unclear Discussion and Unconvincing Conclusions: The paper's discussion lacks clarity, and some conclusions are not well-supported. For instance, the authors claim advantages over TMMLU plus, but only provide a few examples without statistical evidence.
3. Limited Innovation: The benchmark construction and analysis are similar to previous work, and the exclusive use of multiple-choice questions limits innovation. The paper would be stronger with a more novel approach to dataset construction and the inclusion of diverse test types.
4. Data Contamination Concerns: While the authors strive to avoid data contamination, their use of materials from major tests in Taiwan raises concerns. These materials may already be available online in digital form on test preparation websites, potentially accessible to LLMs during training. The observed decline in model performance on older questions (pre-2017) could indicate data contamination, as older materials are less likely to have been digitized and disseminated online.

---

> ### Author Rebuttal · Authors · 2024-05-31
>
> Thanks for the insightful comment! First, we would like to note that [TMMLU-plus](https://arxiv.org/abs/2403.01858) is a concurrent work to ours and hasn’t been published in a peer-reviewed conference.
>
> **Response to RR1**
>
> We would like to emphasize that TMLU’s goal is to foster the development of localized Taiwanese Mandarin LMs. Though academic terminology may be similar to Mainland China, subtle nuances in the way questions are described still capture linguistic variations, which could affect language understanding of localized LMs that are trained only on Traditional Chinese data.
>
> We hypothesize Simplified Chinese LMs have been trained on Traditional Mandarin content, thus, perform sufficiently well. However, at the same parametric scale, Taiwanese Mandarin model Breeze-7B-Instruct greatly outperforms Simplified Chinese model Qwen-7B-Chat (74.35% vs 64.94%) on the Taiwan-specific subset of TMLU (Table 3b).
>
> **Response to RR2**
>
> TMLU offers following distinct features not available in TMMLU-plus:
> 1. We provide clear attribution of the data source (Section 3.2). Also, we suspect TMMLU-plus collects most questions from a single website, which greatly increases the risk of data contamination. To validate this assumption, we conduct additional analysis: Please refer to our response to point 1 for Review 1eqm.
> 2. We manually curated few-shot CoT demonstrations, which is crucial to facilitate the evaluation of reasoning capabilities in Traditional-Mandarin LLMs.
> 3. We standardized all instances containing symbols and equations in LaTEX format for better clarity (Figure 6a).
>
> Collectively, these features make TMLU a more ideal benchmark for the community compared to TMMLU-plus.
>
> **Response to RR3**
>
> Indeed our data construction pipeline is relatively standard, but except the concurrent work TMMLU-plus, there are no MMLU-style benchmarks that are in Traditional Mandarin by construct. We will continue exploring innovative ways to augment TMLU in the future.
>
> **Response to RR4**
>
> We acknowledge that some questions available online might be a part of the training data. Yet, we offer support that TMLU is less likely to suffer from contamination in Section 5., showing that TMMLU-plus perform worse in every tested LMs.
>
> **Response to Q**
>
> Thanks for pointing this out! We will correct the error and replace “Tongzhi” with terms such as “Tudou” (which typically means “potato” in Mainland China and “peanut” in Taiwan).

---

> ### Comment · Reviewer_8Aze · 2024-06-06
>
> Thank you for pointing out that TMMLU-plus is a concurrent work; I have updated the score accordingly.
> However, I am still not entirely convinced by the claim that "TMLU is better for the community to objectively evaluate the model capability due to its better transparency, localization, and robustness." While I admit that TMLU has better data quality, as the authors stated, I am not persuaded regarding its localization and robustness. I noticed that TMMLU-plus includes tests on Taiwanese Trivia and Taiwanese Hokkien, which could cover more local language use, particularly informal language.

---

> > ### Author Response · Authors · 2024-06-07
> >
> > Thank you for the insightful comments! Regarding localization — the Taiwanese Trivia question set (TTQA) included in TMMLU-plus is proposed by another prior work [1]. We are also glad to augment TMLU with TTQA in future revision, along with more questions such as Hokkien knowledge to improve the localization of TMLU. Thanks again for your suggestions.
> >
> > [1] Extending the pre-training of bloom for improved support of traditional chinese: Models, Methods and Results
> > Ennen et al. arXiv:2303.04715, 2023

---

### Official Review · Reviewer_1eqm · 2024-05-14

**Rating:** 6
**Confidence:** 4
**Ethics Flag:** 1

**Summary:**

This paper proposes a benchmark for traditional Chinese, TMLU, which targets the Taiwan society by construction. This benchmark is like the MMLU format but with traditional Chinese language. Compared with previous benchmarks for Taiwanese Mandarin such as TC-Eval and TMMLU, the authors argue that TMLU differs in that it originates from traditional Chinese and it is sourced from PDF or Word documents that suffer from less contamination. The paper benchmarks many LLMs on the proposed TMLU benchmark.

**Questions To Authors:**

```
Understanding that TMMLU-Plus is a concurrent work within one month of the submission, I increase my score from 4 to 6 given that I ignore the existence of TMMLU-Plus
```

**Reasons To Accept:**

1. Creating an MMLU-like benchmark for Taiwanese Mandarin is a focused contribution and could benefit the deployment and development of LLMs to serve Taiwan users.
2. Empirical experiments are comprehensive, there are COT and data contamination analysis included besides the main results.

**Reasons To Reject:**

1. While the authors try to distinguish TMLU from TMMLU-plus, it is better to have more quantitative evidence given that the two benchmarks are similar. For example, Table 1 notes that the robustness, transparency of TMMLU-plus are low, what do they mean? In Section 2 the authors mention that “In our preliminary investigation, we found that most questions in TMMLU-plus could be found on one single website.”, can you quantify this? How many can be found?
2. In figure 3, I wonder what the absolute values of the y-axis mean, for example, does a min-k% prob of 4.0 indicate that the pretraining data is absolutely likely to present in the pretraining data? Currently the comparison in Figure 3 is relative, and I am not sure whether the gap between TMLU and TMMLU-plus is significant or has meaningful differences given that the gap looks small – maybe TMLU and TMMLU-plus both suffer from data contamination already.

---

> ### Author Rebuttal · Authors · 2024-05-31
>
> Thanks for the thorough review! First, we would like to note that [TMMLU-plus](https://arxiv.org/abs/2403.01858) is a concurrent work to ours and hasn’t been published in a peer-reviewed conference.
>
> **Response to RR1**
>
> The robustness of TMMLU-plus is considered relatively low comparing to TMLU because of the following points discussed in Section 2:
> 1. Most questions in TMMLU-plus could be found on one single website, greatly exposing its risk to test data contamination if the website is scrapped for pre-training.
> 2. TMMLU-plus contains unanswerable questions that require information (e.g., images or tables) not accompanied in the dataset (Figure 6b).
>
> Table 1 is a concise comparison which summarizes the discussion in Section 2. We will strengthen the description in Section 2 to outline a clear connection to Table 1.
>
> Regarding our claim – “we found that most questions in TMMLU-plus could be found on one single website.”, we conduct additional evaluation to provide more quantified evidence.
> 1. We randomly sample 100 instances from TMMLU-plus and discover 91 of them are on the website: https://app.yamol.tw/ which we suspect to be the source of TMMLU-plus (TMMLU-plus does not offer clear information about their source — they simply mention “standardized test resources and PDF documents available online” in Section 2.2 in their preprint).
> 2. We examine the full mC4 dataset and find 28,685 entries have URLs from the suspected website: https://app.yamol.tw/.
>
> **Response to RR2**
>
> Thanks for the insightful question! The comparison metric in Figure 3, Min-K%, is an indicator of how likely the given text is in the pre-training data of the examined LM. Min-K% is by nature relative, where its absolute value varies under different settings (e.g., model size, vocabulary size).
> Note that it requires oracle labels of the tested dataset to determine a threshold that reflects whether contamination occurs (Please refer to Section 5.1 in the Min-K% paper).
>
> We acknowledge the possibility that both TMLU and TMMLU-plus suffer from data contamination. Nonetheless, we can observe a systematic and unifying behavior in Figure 3, which shows that TMLU is less likely to be contaminated compared to TMMLU-plus in every model tested.

---

> > ### Comment · Reviewer_1eqm · 2024-06-04
> >
> > Thank you for the clarifications! Understanding that TMMLU-plus is a concurrent work, I think the contribution of this work is more significant. However, I do suggest that when discussing some drawbacks of TMMLU-plus, it is more convincing to use objective evidence than subjective judgements, for example, if we talk that TMMLU-plus contains questions that require additional images/tables, it is advised to report a percentage maybe within only 50 examples, otherwise this point is not very meaningful.
> > I increase my score to 6 given that I ignore the existence of TMMLU-plus.

---

> > > ### Author Response · Authors · 2024-06-07
> > >
> > > Thanks for the constructive feedback! We will strengthen our objective evidence by providing more relevant samples in the final version.

---

### Decision · Program_Chairs · 2024-07-10

**Decision:**

Accept

**Comment:**

This paper introduces a benchmark to evaluate the performance of language models in understanding Taiwanese Mandarin, which is an important task, as all reviewers (including me) found, to improve inclusivity with the rapid development of large language model techniques, as well as evaluating a number of models on the proposed benchmark.
I resonate with Reviewer 8Aze that, in principle, it would be good to include some diagnostic test cases that better represent the linguistic and/or cultural differences between Chinese and Taiwanese Mandarin, especially when adapting a Simplified Chinese model.
That being said, the benchmark is still very valuable for localizing language models into Taiwanese Mandarin.

Please also follow the suggestions by Reviewers u2Ve and 8Aze to correct the Chinese Mandarin terms with misunderstanding.